# Easily Synthesized Polyaniline@Cellulose Nanowhiskers Better Tune Network Structures in Ag-Based Adhesives: Examining the Improvements in Conductivity, Stability, and Flexibility

**DOI:** 10.3390/nano9111542

**Published:** 2019-10-30

**Authors:** Ge Cao, Xiaolan Gao, Linlin Wang, Huahua Cui, Junyi Lu, Yuan Meng, Wei Xue, Chun Cheng, Yanhong Tian, Yanqing Tian

**Affiliations:** 1School of Materials Science and Engineering, Harbin Institute of Technology, Nangang District, Harbin 150001, China; 11749286@mail.sustc.edu.cn (G.C.); 11849284@mail.sustech.edu.cn (W.X.); 2Department of Materials Science and Engineering, Southern University of Science and Technology, Xili, Nanshan District, Shenzhen 518055, China; gaoxl@mail.sustc.edu.cn (X.G.); 11749154@mail.sustech.edu.cn (L.W.); cuihh3@mail.sustech.edu.cn (H.C.); 11612916@mail.sustech.edu.cn (J.L.); 11611831@mail.sustech.edu.cn (Y.M.); chengc@sustc.edu.cn (C.C.)

**Keywords:** electrically conductive adhesives, preparation, polyaniline@cellulose, electrical properties, flexible electronics

## Abstract

It is essential to develop a novel and versatile strategy for constructing electrically conductive adhesives (ECAs) that have superior conductivity and high mechanical properties. In this work, easily synthesized polyaniline@cellulose (PANI@CNs) nanowhiskers with a high aspect ratio and excellent solubility in 1,4-dioxane were prepared and added to conventional Ag-containing adhesives. A small amount of PANI@CNs can dramatically tune the structure of the ECAs’ conductive network and significantly improve the conductivity of the ECAs. Good solubility of PANI@CNs in solvents brings excellent dispersion in the polymer matrix. Thus, a three-dimensional (3D) conducting network formed with dispersed PANI@CNs and Ag flakes can enhance the conductivity of ECAs. The conductivity of the ECAs (with 1.5 wt% PANI@CNs and 55 wt% Ag flakes) showed three orders of magnitude higher than that of the ECAs filled with 55 wt% Ag flakes and 65 wt% Ag flakes. Meanwhile, the integration of PANI@CNs with Ag flakes in polymer matrices also significantly enhanced the mechanical compliance of the resulted ECAs. The resistivity remained unchanged after rolling the PANI@CNs-containing ECAs’ film into a 4 mm bending radius for over 1500 cycles. A bendable printed circuit was fabricated using the above PANI@CNs-containing ECAs, which demonstrated their future potential in the field of flexible electronics.

## 1. Introduction

The increasing demand for consumer electronics where circuits are wrapped or rolled up conformably around complex shapes has prompted significant interests in next-generation interconnect materials [1]. Electrically conductive adhesives (ECA) has been considered the most promising interconnects to substitute lead/tin solder for flexible electronics [2,3]. In comparison with long-established Sn–Pb based eutectic solders, ECAs have numerous advantages, such as low processing temperature for interconnecting the fragile integrated circuits, toxic-free, and high safety for environment system [4,5,6]. Even though ECAs have many merits, they still have a long way to go to perfectly meet the needs of today’s flexible electronic products. The main obstacle is that current flexible ECAs usually have limited conductivity (resistivity is usually higher than 2.0 × 10^−4^ Ω·cm) or need significant amounts of metal fillers to achieve reasonable conductivity. These limit the widespread applications of ECAs in microelectronics [7,8]. Therefore, it is imperative to enhance the conductivity of ECAs with a minimized amount of metal fractions to reduce the cost and prevent brittle fracture of the ECA’s patterns upon mechanical deformation [4,9].

Generally, generating three-dimensional (3D) conducting networks with high quality among metals and other fillers in the matrix could be helpful for improving the electrical conductivity of ECAs while using minimal metal volumetric fractions. For conventional Ag-containing ECAs, the conductivity was significantly improved by adding high-aspect-ratio conductive materials such as carbon nanotubes (CNTs) [10], graphene [11], and silver nanowires (Ag NWs) [12]. These studies showed that the conductivity of ECAs is responsive to the high-aspect-ratio conductive materials in the composites containing hybrid fillers with different dimensions. The electrical conductivity of these composites with hybrid fillers is often substantially higher than the average value of those only contain silver flakes alone, indicating that a considerable amount of expensive metal fillers could be substituted with high-aspect-ratio cheap conductive fillers. These high-aspect-ratio fillers can act as bridges to connect neighboring silver flakes, which can accelerate electron-transporting. These results indicated that the total cost of the adhesives could be reduced by this method.

Some ECAs that contain different dimensions of conductive fillers have also been reported [13,14,15,16,17]. Many different conductive enhancers were used to construct hybrid ECAs to improve their electrical conductivity, such as metal nanoparticles, metal nanowires, or carbon nanomaterials. However, there are some drawbacks to these materials. For example, metal nanoparticles and nanowires were very expensive, and their surface was always covered by insulating organic molecules, which significantly weakened the electrical conductivity of ECAs [18,19]. Owing to their hydrophobic surfaces and large specific surface area, carbon-based nanofillers such as CNTs and graphene have poor dispersion in the solvents for composites, which generally require surface chemical modification or dispersing agent to reach a more stable suspension [20,21].

Along with the continuous advancement of conductivity enhancer, conducting polymers have attracted extensive interests due to their relatively low cost of monomers and excellent chemical stability, ease of synthesis, and high conductivity properties [22,23]. Tian et al. [24] reported a kind of modified polypyrrole where nanoparticles were added to epoxy-based ECAs with Ag fillers in order to enhance ECA conductivity. Zhao et al. [25] fabricated a hybrid ECAs that contained Ag flakes and PEDOT: PSS nano-gels. Their results showed that the combination of an optimum 1 wt% PEDOT: PSS in conventional Ag-containing ECAs (60% wt Ag contents) significantly improved the conductivities by 25 times with negligible impact on the adhesion of the ECAs.

Despite the success in the development of conducting polymers for conductivity enhancing materials, some problems still exist that hinder the broad commercial applications of these composite ECAs materials. The principal obstructions for advanced ECAs that use polymers are: (i) the conductive polymers currently used are typically in the format of spherical particles that limit the ability to form high-quality three-dimensional conductive network structures in these conductive pastes, thus impeding the improvement of conductivity; and (ii) most conductive polymers have relatively weak processing capabilities due to their insolubility in conventional solvents [26,27], which severely hampers their further application.

In order to alleviate the above two problems, we report a facile method to prepare polyaniline (PANI)/cellulose nanowhiskers (PANI@CNs) as novel nanocomposites that possess a high aspect ratio (defined as the ratio of length to diameter; the aspect ratios of PANI@CNs were about 10–15) and excellent solubility in 1,4-dioxane (nearly 5 mg/mL). Cellulose nanowhiskers (CNs) are of the most promising biomaterials derived from natural cellulose fibers [28]. They have received extensive interest due to their distinctive characteristics like high-aspect-ratio, regulatable morphologies, and excellent dispersity in conventional solvents [29,30]. CNs characteristics make cellulose nanowhiskers excellent templates to support the stabilization of PANI with a high aspect ratio and high conductive composites. A large number of researches have been devoted to the synthesis of a wide variety of cellulose-based conductive polymer materials, which are widely used in the fields of supercapacitors, electromagnetic shielding, and drug release [31,32,33,34]. However, to the best of our knowledge, the introduction of PANI@CN as a way to reach superior electrical conductivity with the low fabricating cost of conductive adhesives has not been reported. In this study, a layer of conductive PANI was polymerized in situ on CNs templates to construct high-aspect-ratio conductive fillers. The formed PANI@CNs nanocomposite with Ag flakes was then dispersed into thermoplastic polyurethane (TPU) resin to fabricate hybrid ECAs by a simple solution mixing method. These PANI@CNs nanocomposites are expected to locate in the interfaces between adjacent Ag flakes and exhibit 3D hierarchical conductive networks for improving the conductivity of the ECAs. Therefore, new flexible conductive adhesives with ultrahigh conductivity and high mechanical compliance were achieved.

## 2. Experimental Section

### 2.1. Experimental Materials

Ammonium persulfate (APS), Aniline (ANI), and 1,4-Dioxane were offered using Adamas Reagent Co., Ltd. (Shanghai, China). Sulfuric acid (H_2_SO_4_) was obtained from Shenzhen Chemical Reagent Technology Co., Ltd. (Shenzhen, China). Thermoplastic polyurethane (TPU) was purchased from Dongguan Guangye Plastic Materials Co., Ltd. (Dongguan, China). Silver micron-flakes were provided from Shanghai Xinzhuan Alloy Materials (Shanghai, China). Medical purified cotton was obtained by Guangzhou Yulongge Biological Technology Co., Ltd. (Guangzhou, China). Light-emitting-diode (LED) chips and resistors were offered by Changzhou Keyun Electronics Co., Ltd. (Changzhou, China).

### 2.2. Synthesis of Cellulose Nanowhiskers (CNs)

Similar to Lu’s previous studies [35], CNs were obtained by acid treatment of medical purified cotton. The cotton was cut into small pieces; these small pieces were then mixed with sulfuric acid under strong agitating at 25 °C for 120 min. Later, this reaction was quenched by a large amount of deionized water. The suspension was centrifuged, and then the solid was dialyzed several times in water until the pH of the suspension was monitored to be neutral. The obtained acid-treated cellulose nanowhiskers were ultra-sonicated at 25 °C for 1 h. Finally, the suspension was concentrated to be 0.5 wt% by a rotary evaporator for later use.

### 2.3. Synthesis of PANI@CNs Nanowhiskers and PANI

The experimental strategy for the synthesis of PANI@CNs nanowhiskers was illustrated in Figure 1. Aniline (ANI) was physically adsorbed on the surface of CNs. The oxidant (APS) was then introduced into the mixture to polymerize aniline. A typical procedure described below: ANI solution (0.1 mol/L) was prepared by dissolving aniline in dilute H_2_SO_4_ (1 mol/L). The solution was then homogeneously mixed in a previously prepared CN suspension. The obtained mixture was stirred at 25 °C for 10 min. APS dissolved in dilute sulfuric acid (1 mol/L) solution was poured into the reaction mixture under vigorous agitation to trigger polymerization of ANI. After two hours, a deep green suspension appeared. The obtained product was filtered and rinsed three times. By adjusting the ratios of aniline to CNs, the PANI@CNs nanowhiskers with different CNs/aniline weight ratios were successfully synthesized. As a reference, conventional PANI was obtained by the identical method except that no CNs was added the reaction mixture.

### 2.4. Preparation of ECAs

The ECAs were fabricated based on our previously reported procedure with minor modifications [36]. A typical procedure was given below: 0.03 g PANI@CNs nanowhiskers were well dispersed in 1,4-dioxane. Subsequently, 0.5 g TPU particles were dissolved into the above suspension. Then, we added 0.63 g silver powder to the mixture under gently agitating for about 2 h. The resulted mixture was heated in air at 80 °C to remove 1,4-dioxane. The TUP/PANI@CNs ECAs’ films were prepared via the solvent-evaporation method.

### 2.5. Characterization and Measurements

The morphologies of the PANI@CNs nanowhiskers, silver flakes, and the ECAs were characterized by scanning electron microscope (SEM, MIRA3, TESCAN, Brno, Czech).

The conductivity of PANI@CNs and ECA samples was determined with a four-point probe system (RTS-9, 4 PROBES TECH, Guangzhou, China). The detailed measurement process is consistent with the literature we previously published [32].

The current-voltage characteristics of the pattern of different ECAs were determined with an electrochemical workstation (Ametek Parstat, 3000A-DX, Berwyn, IL, USA).

The chemical structure of PANI@CNs nanowhiskers, PANI were determined with an FT-IR ( Fourier Transform Infrared Spectroscopy) characterization by a Nicolet 6700 spectrophotometer (Waltham, MA, USA). The samples were blended with KBR powder to compress into tablets for FT-IR measurement (range of 4000 to 400 cm^−1^).

The thermodynamic stability of PANI@CNs nanowhiskers, PANI was measured with a METTLER TGA-1 (Mettler, Columbus, OH, USA) at a heating rate of 5 °C/min.

The bending and recording resistance changes of the ECAs were employed using a specialized stretching-system, which was given in the support information (Appendix A).

## 3. Results and Discussion

### 3.1. The Preparation of ECAs

Figure 1 shows a general strategy for preparing PANI@CNs/silver-PU-ECAs adhesives. The PANI@CNs composites were obtained using a polymerization process of aniline with APS as an initiator and CNs as a template. The obtained PANI@CNs were dispersed into the TPU matrix with silver micron-flakes (the detailed information in Appendix A) to construct the PANI@CNs/silver-PU-ECAs adhesives. Detailed compositions for the prepared ECAs with different compositions were given in Table 1.

### 3.2. Solution Property and Morphology of CNs

Figure 2a showed the photo of a bottle of pure cellulose solution (1 wt%). The solution looks almost transparent. Due to the surface charge, CNs are very stable in water. The solution has high suspension stability of more than three months. Atomic force microscopy (AFM) images in Figure 2b showed that the CNs exhibited rod-shaped morphologies with a relatively uniform diameter distribution. The diameters of rod-like celluloses range from 10 to 35 nm. Their lengths are about 100−300 nm. The results showed that cellulose nanowhiskers were successfully prepared.

The morphologies of PANI@CNs with different CNs/aniline weight ratios are given in Figure 3. It was found that when there is no cellulose in the reaction mixture, the obtained polyaniline is in the form of globular particles (Figure 3a). When cellulose was gradually added in the reaction mixture, PANI@CNs nanocomposites that possessed rod-like shapes were obtained. When the weight ratio of the CNs/aniline weight ratio is 1/6, the small number of cellulose nanocrystals is difficult to adequately react with the excess of the aniline monomer, resulting in the formation of irregular complexes and a large amount of PANI nanoparticles (Figure 3b). By increasing the CNs mass ratio to 1:4, more uniform PANI@CNs nanowhiskers with thicker shells formed. Moreover, the nanocomposites are well dispersed and exhibited a short rod-like shape (Figure 3c). Aggregations of PANIs on a large scale were not observed. When the ratio of CNs/aniline further increased to 1:2, PANI@CNs nanowhiskers did not change obviously (Figure 3d). However, it is worth noting here that the resistivity of PANI@CNs nanowhiskers increased because of the higher proportion of non-conductive CNs, which will be discussed in the later section.

### 3.3. Chemical Structure Characterization

The structure of the CNs, PANI, and PANI@CNs (with the weight ratio CNs/aniline of 1:4 and 1:2) was examined systematically by means of FT-IR analysis. For the FT-IR spectra of CNs, the characteristic peak of O-H stretching was observed in the vicinity of 3400 cm^−1^. The IR band near 2900 cm^−1^ was ascribed to the asymmetric vibration of C-H. The peak at 1659 cm^−1^ is due to the H-O-H bending produced by the water of CNs. The characteristic band at 1164 cm^−1^ corresponds to the C-O antisymmetric bridge stretching. The influential band at 1061 cm^−1^ is caused by the vibration of the C-O-C pyranose ring skeleton. These IR bands represent the usual cellulose IR band, as reported in previous literature [37,38].

The IR bands about 1500 and 1580 cm^−1^ were ascribed to the stretching vibrations of the N-benzopyrene-N and N = hydrazine moieties = N of PANI. The IR band around 1130 and 1290 cm^−1^ were noted due to the vibration of C-H (benzene ring) and the C-N bond. Other vibration of the C-H (para-disubstituted benzene) emerged around 800 cm^−1^. These IR bands are also similar to the conventional PANIs reported by Stejskal et al. before [39].

For the FT-IR spectra of PANI@CNs, almost all of the IR-band of the spectra of PANI and CNs could be observed. These observations indicated that PANI formed successfully on the CNs surface during the in-situ polymerization. Moreover, the intensity of the IR-band of PANI@CNs near 3420 cm^−1^ increased significantly compared to that of PANI, indicating that there are more hydroxyl groups on PANI@CNs. This property might be helpful for the hybrids to form homogeneous dispersion in aqueous solution.

TGA (Thermal gravimetric analysis) curves of CNs and PANI@CNs and PANI were displayed from 25 to 600 °C (Figure 4b). All TGA curves can be divided into three sections. (1): At the beginning section from 25 to 120 °C, the water evaporated in these samples to result in slight weight loss; (2): The primary weight loss of CNs occurred in the middle section from 120 to 350 °C. A considerable loss of PANI and PANI@CN were observed around 120 to 400 °C. Compared with CNs (35.3% residual), PANI and PANI@CNs (with the weight ratio CNs/aniline of 1:4 and 1:2) showed higher residues: 58.2%, 50%, and 45%, respectively. The results showed the thermal stability of PANI@CNs exceeds that of CNs and is slightly lower than that of PANI. With the increase of CNs, the thermal stability of PANI@CNs decreased, making the processing at high temperature difficult. Thus, such new materials are suitable for low-temperature applications from room temperature to 200 °C, not suitable for high-temperature applications.

### 3.4. The Conductivity and Solution Dispersion of PANI and PANI@CNs

To highlight the vital role of CNs, we tested the conductivity and solution dispersion of PANI and PANI@CNs.

Benefiting from the high-aspect-ratio morphologies that provided shorter contact distance, PANI@CNs nanowhiskers showed greatly enhanced conductivity compared to PANI with typical spherical morphologies. When the CNs/aniline weight ratio of 1:4, the PANI@CNs with one-dimensional extended morphologies (Figure 3c) had the conductivity of 1.64 S cm^−1^, which was almost three times that of its globular counterpart with 0.53 S cm^−1^. When more CNs were added, the conductivity of the composites came down due to the higher proportion of non-conductive CNs. For the CNs/aniline weight ratio increased to 1:2, the PANI@CNs nanowhisker had a conductivity of 0.41 S cm^−1^, which was slightly lower than that of its globular counterpart (Figure 5a).

After mixing with 1,4-dioxane (for dissolving TPU) and being ultrasonically treated, the obtained PANI@CNs suspension exhibited excellent dispersibility. After an ultrasonicated treatment, PANI@CNs nanocomposites (5 mg/mL) were formed as a stable colloid suspension in 1,4-dioxane. Any aggregates were not identified in the suspension, while tiny aggregates of PANI were noticeable in the PANI only suspension (5 mg/mL) (Figure 5b). In addition, we further tested the dispersion of PANI@CNs in water and ethanol. The results (Appendix A) show that the PANI@CNs show excellent dispersity in these two green solvents, which may provide a choice for fabricating environmentally friendly high-performance ECAs in the future.

From the above tests, it can be concluded that cellulose nanocrystals have at least two critical functions in the synthesis process. (i) CNs provided a one-dimensional template that promotes a favorable growth of PANI on CNs surface, which could connect neighboring nanotubes. These high- aspect-ratio nanotubes might be more likely to form conductive networks than spherical nanoparticles. Thus, a low percolation threshold could be achieved in polymer matrices with nanotubular fillers [40]. (ii) Secondly, CNs as macromolecules with productive hydroxyl groups can act as excellent dispersants to reduce the aggregations of PANI@CNs nanowhiskers to form a stable colloid suspension in aqueous solution.

### 3.5. Bulk Resistivity and the Conducting Mechanism of the ECAs

The bulk resistivity of sequences of hybrid adhesives with different silver flakes loading was given in Figure 6a. The resistivity of the ECAs reached 3.16 × 10^−5^ Ω·cm at 55 wt% Ag content and 1.5 wt% PANI@CNs after cured at 25 °C for 120 min and cooled to r.t. for measurement. The resistivity of PC-ECAs (1.5 wt% PANI@CNs and 55 wt% silver content, designated PC-1, Table 1) is 1/1000 of that of PU-ECAs filled 55 wt% Ag flakes exclusively, which is surprisingly 1/200 of PU-ECAs containing 65 wt% Ag flakes (PU-2). Additionally, the current-voltage characteristics of the pattern of different ECAs are shown in Appendix A. Compared with the conventional Ag-containing ECAs’ pattern (PU-1, PU-2), the hybrid ECAs (PC-1) could be found to exhibit ultra-low resistance. These data indicated that the addition of a small amount of PANI@CNs nanowhiskers to conventional silver adhesives could significantly reduce the adhesive’s electrical resistance.

For the PC-ECAs, an ultrahigh conductivity was acquired from the addition of PANI@CNs nanowhiskers instead of the use of many high-price silver flakes. The superior electrical performance of the ECAs could be attributed to two critical factors. The first factor is that various kinds of conductive enhancers have powerful synergistic effects on elevating the electrical performance of the hybrid ECAs. The utilization of the high-aspect-ratio PANI@CNs nanowhiskers changed the conductive network structure of the ECAs. PANI@CNs nanowhiskers with Ag flakes constructed 3D conductive networks to achieve efficient interconnection among Ag flakes in the polymer matrix. SEM images showed the conductive network consisting of silver flakes and PANI@CNs nanowhiskers in the PC-ECAs (Figure 6b). The graphics displayed that PANI@CNs nanowhiskers have no visible aggregations in the polymer matrix. PANI@CNs were well dispersed among the silver flakes and built a 3D conductive network. The data in Figure 6c also supported the above consideration by showing that the conductivity of PC-ECAs is much higher than those of binary ECAs filling conventional PANI particles. The enhanced properties were attributed to high-aspect-ratio PANI@CNs nanowhiskers (the aspect ratios were about 10–15, Appendix A) were more comfortable to act as bridges among the silver flakes to form 3D conducting networks than spherical PANI nanoparticles.

The second factor is the selection of a novel solution-mixing method for fabricating ECAs. Different from the insolubility of conventional PANI nanoparticles in common solvents, PANI@CNs nanowhiskers have very high solubility in 1,4-dioxane (that could effectively dissolve TPU resin), which makes the possibility of choosing the solution-mixing method for constructing ECAs. The solution-mixing method included three main sections for fabricating PC-ECAs: (1) dispersing PANI@CNs in 1,4-dioxane to form a stable suspension, (2) blended TPU pellets and Ag flakes in suspension, and (3) preparing PC-ECAs by the solvent evaporation method. Compared to the melt blending method (usually for constructing conductive adhesives), which can only achieve macroscopic dispersion [41], the solution-mixing method for preparing ECAs could bring better fillers’ dispersion due to hardly need of the consideration of the polymers’ viscosities.

It is important to note that PC-ECAs (PC-1) exhibit ultrahigh electrical conductivity even prepared by room temperature. In Figure 6c, the hybrid ECAs shows resistivity of 3.16 × 10^−5^ Ω·cm when cured at room temperature. For comparison, many conventional Ag-based adhesives still needed a much high processing temperature to obtain good electrical conductivity. Sometimes, for the purpose of triggering the sintering of silver in ECAs at low temperatures, very tiny and expensive metal nanocrystals were required to add into conventional Ag-based adhesives. Even so, in many cases, the sintering temperature of these conductive pastes is still above one hundred degrees [42,43]. In Figure 6c, at low processing temperatures (25 °C–150 °C), the resistivity of the hybrid conductive paste (PC-1) changed slightly with increasing temperature. This indicated that the hybrid ECAs maintained good electrical conductivity at low temperatures, which is imperative for developing suitable packaging and sealing materials for protecting flexible electronics circuits.

### 3.6. The Conductivity of PC-ECAs under Mechanical Deformation

Flexible electronics have attracted considerable attention from scientists over the past two decades. In order to match flexible devices, ECAs need to maintain electrical performance under mechanical deformation. Figure 7 shows the change in resistance of the PC-ECAs film (PC-1) on a PET substrate under high bending deformation. Figure 7a shows the relationship between the normalized resistance (R/R_0_) and the bend radius of the ECAs. Almost any resistance changes were observed after rolling the films into a bending radius of 4mm. This stability far exceeds epoxy-based silver adhesives, and even more than Ag-carbon nanotube/silver pastes [4]. Furthermore, the variations of R/R_0_ are less than 5% after 1500 cycles. Figure 7b,c shows detailed information on the variations in the resistance of the film on the PET substrate during the cycle from the 1114th to the 1120th. It is worth noting that the resistance changes of every cycle are less than 2%, which indicates that PC-ECA has excellent electrical stability under high mechanical deformation. Figure 7d,f showed the PC-ECAs before and after 1500 times bending. Any noticeable crack was not observed after the 1500 bending. Furthermore, PC-1 films still maintained their excellent electrical conductivity, as shown in Figure 7e.

### 3.7. Demonstration of the in Flexible Electronics Applications of the PC-ECAs

LED chips and resistors were connected to the PC-ECAs (PC-1) patterns on a flexible PET for demonstrating the use of the adhesives in printed electronics applications. Figure 8a,b showed printed patterns that bent easily made by PC-ECAs. A printed flashing circuit was displayed in Figure 8c. The lightness of these LED chips barely changed obviously when the device was under highly flexural states. (Figure 8d,e) It shows that the ECAs’ circuit has superb electrical stability under high distortion. This might offer a novel approach to connect flexible electronics.

### 3.8. Comparison of the Electrical Performances of the PC-ECAs and the Literature ECAs

As is listed in Table 2, the conductivities of PC-ECAs (PC-1) were compared with other literature ECAs. It was found that our PC-ECAs (PC-1) filled 55 wt% Ag contents has an extremely low resistivity of 3.16 × 10^−5^ Ω·cm. By comparing our data with recent work in Table 2, it clear that the electrical performances of PC-ECAs with little Ag content surpassed other literature ECAs. These properties of PC-ECAs could contribute to the goal of improving the conductivity of ECAs while reducing the proportion of high-priced metal fillers used.

## 4. Conclusions

In summary, we reported a simple method to prepare a kind of high-aspect-ratio PANI@CNs nanowhiskers with good solubility in 1,4-dioxane. These high-aspect-ratio PANI@CNs were added to conventional Ag-containing adhesives in order to fabricate flexible ECAs with high-performance. The ECAs offered extremely low electrical resistivity (3.16 × 10^−5^ Ω·cm) and contained only 55 wt% of silver contents at room temperature. Moreover, they were cured and kept reasonably stable in a highly mechanical deformation. Further, there was a 5% resistance change after rolling the ECAs’ film to a 4 mm bending radius after 1500 cycles. The combination of these properties will make the new PC-ECAs a very promising interconnect material for flexible and printed electronics.

## Figures and Tables

**Figure 1 nanomaterials-09-01542-f001:**
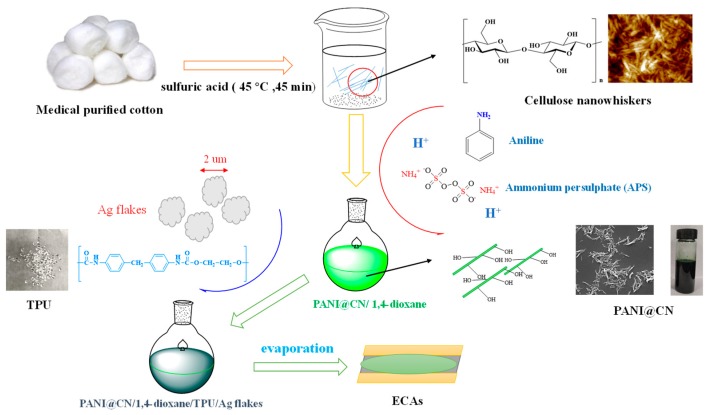
Schematic illustration of the preparation of polyaniline@cellulose (PANI@CNs)/silver-PU-electrically conducted adhesives (ECAs).

**Figure 2 nanomaterials-09-01542-f002:**
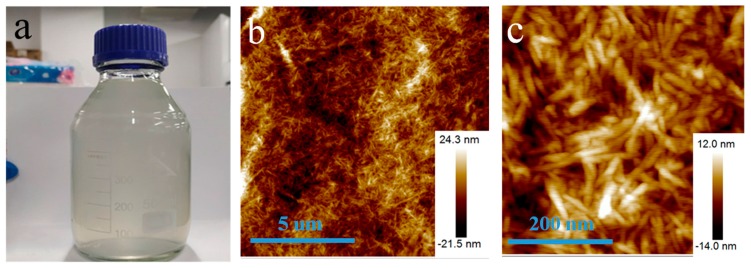
(**a**) The photo of a bottle of CNs aqueous dispersion, (**b**) a low magnification AFM image of CNs, and (**c**) a high magnification AFM image of CNs.

**Figure 3 nanomaterials-09-01542-f003:**
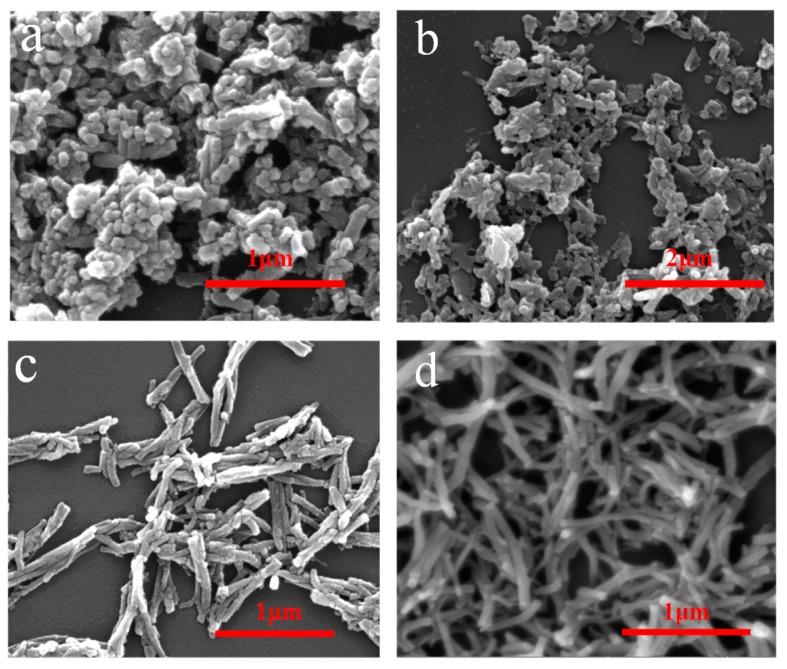
Scanning electron microscope (SEM) images of PANI (**a**), PANI@CNs nanowhiskers with the weight ratio CNs/aniline (1:6) (**b**), (1:4) (**c**), and (1:2) (**d**).

**Figure 4 nanomaterials-09-01542-f004:**
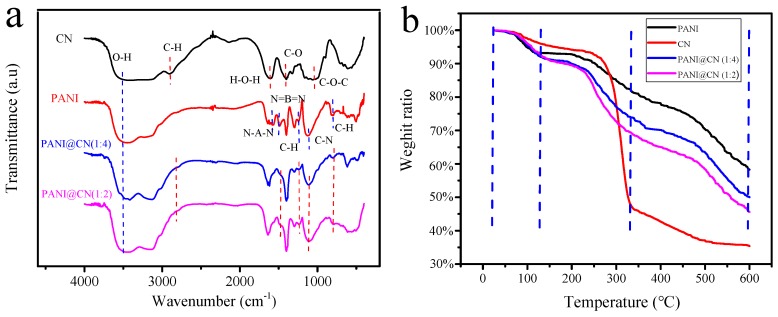
(**a**) FT-IR spectra of CNs, PANI, and PANI@CNs with the different weight ratios of CNs/aniline; (**b**) TGA curves of CNs, PANI@CNs, and PANI.

**Figure 5 nanomaterials-09-01542-f005:**
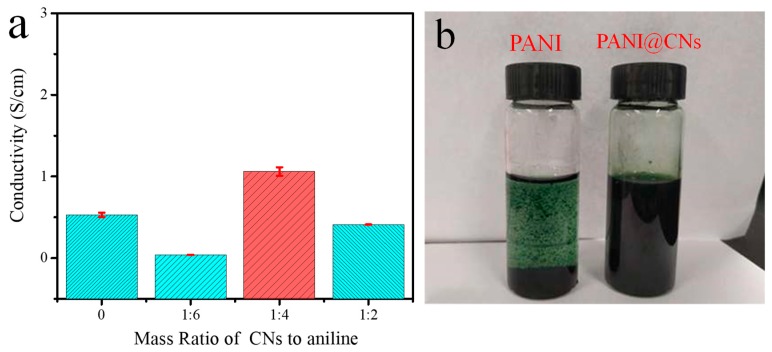
(**a**) Electrical conductivity of PANI@CNs with different feeding mass ratios of CNs/aniline, (**b**) dispersion test of PANI, and PANI@CNs (1:4) in 1,4-dioxane.

**Figure 6 nanomaterials-09-01542-f006:**
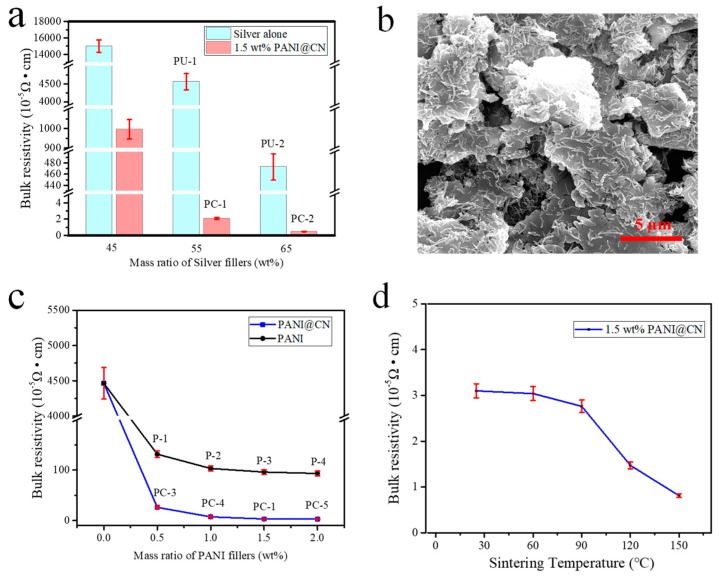
(**a**) Bulk resistivity of the ECAs with different silver content, (**b**) SEM images of PC- -1ECAs, (**c**) bulk resistivity of hybrid ECAs containing 55 wt% Ag fillers (PC-1, 3, 4, 5; P-1, 2, 3, 4) as a function of different PANI and PANI@CNs, and (**d**) the hybrid ECAs (PC-1) curing at different temperatures.

**Figure 7 nanomaterials-09-01542-f007:**
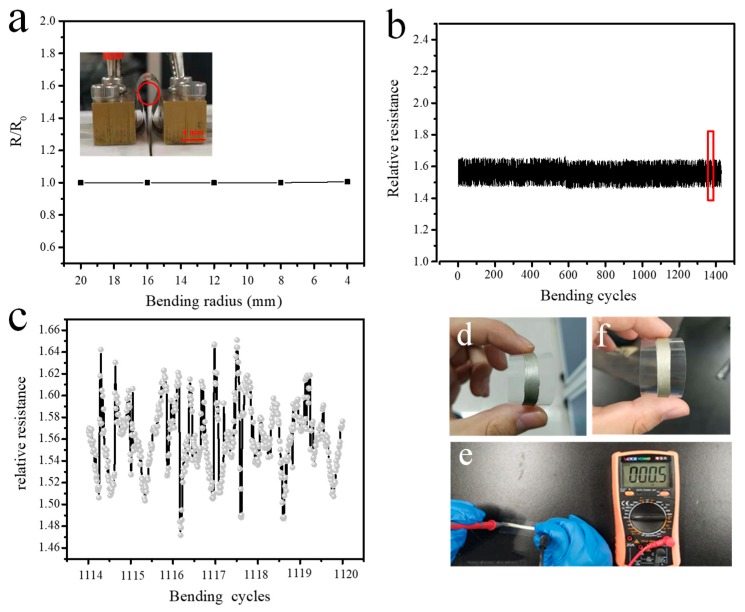
The electrical resistivity of PC-ECAs’s patterns (PC-1) as a functional of rolling radius (**a**), the cyclic performance of PC-ECAs s patterns (**b**), the resistance change of the patterns on the 114–1120 bend-release cycles (**c**), the patterns before 1500 bending cycles (**d**), the patterns after 1500 bending cycles (**f**), and the electrical resistance of the patterns after 1500 bending cycles (**e**).

**Figure 8 nanomaterials-09-01542-f008:**
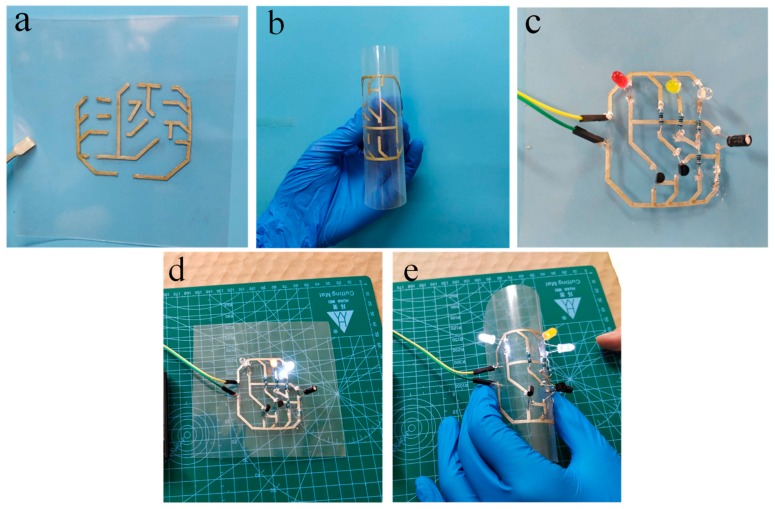
Application of PC-ECAs (PC-1) in printed circuits. (**a**) Printed circuits with line width of 1mm, (**b**) printed circuit at a flexural state, (**c**) a flashing LED device on the printed circuit, (**d**) a flashing LED device working under the flat state, and (**e**) the lightness does not change when the devices are close to ninety degrees bending.

**Table 1 nanomaterials-09-01542-t001:** The compositions for the prepared ECAs.

Sample Code	TPU Resin (wt%)	Ag (wt%)	PANI@CNs (wt%)	PANI (wt%)
PC-1	43.5	55	1.5	0
PC-2	33.5	65	1.5	0
PC-3	44.5	55	0.5	0
PC-4	44	55	1	0
PC-5	43	55	2	0
PU-1	45	55	0	0
PU-2	35	65	0	0
P-1	44.5	55	0	0.5
P-2	44	55	0	1
P-3	43.5	55	0	1.5
P-4	43	55	0	2

Note: the hybrid adhesives containing Ag flakes and PANI@CNs were named as PC series; the adhesives only containing Ag flakes were named as PU series; the hybrid adhesives containing Ag flakes and PANI particles named P series.

**Table 2 nanomaterials-09-01542-t002:** The comparison of conductivities of literature ECAs.

	Content	Curing Temperature (°C)	Electrical Resistivity (10^−5^ Ω·cm)
This work	55 wt% Ag flakes, 1.5 wt% PANI@CNs, 43.5 wt% TPU	25	3.16
Li et al. [4]	80 wt% Ag flakes, 20 wt% PU	180	1
Fu et al. [13]	30.0 wt% Ag flakes, 2.5 wt%Ag naonparticles 7.5% Ag nanowires, 60 wt% epoxy	150	19.6
Yao et al. [8]	70 wt % KI treated Ag, 30 wt% epoxy resin	150	10.8
Yao et al. [10]	50 wt %Ag, 4.5 wt% CNTs, 55.5 wt% PU	120	270.27
Amoli et al. [44]	80 wt% Ag flakes, 1.5 wt% SDS-stabilized graphene nanosheets, 18.5 wt% epoxy resin	150	1.6
Zhang et al. [45]	80 wt% of 4:6 molar ratio of Ag nanoparticles and Ag flakes, 20 wt% epoxy resin	230	0.6
Zhang et al. [14]	75 wt% of 2:3 Ag nanowires and Ag flakes, 25 wt% epoxy resin	300	0.58
Wen et al. [24]	70 wt% Ag flakes, 2.5 wt% polypyrrole nanoparticles, 27.5 wt% epoxy resin	160	9.4
Wen et al. [46]	65 wt% Ag flakes, 0.5 wt% PANI particles, 34.5 wt% PU	25	14.5

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
