# Peer review of "Easily Synthesized Polyaniline@Cellulose Nanowhiskers Better Tune Network Structures in Ag-Based Adhesives: Examining the Improvements in Conductivity, Stability, and Flexibility"

_nanomaterials, 2019, doi:10.3390/nano9111542_

Round 1

Reviewer 1 Report

The manuscript of Cao et al. describes the preparation of flexible silver adhesive through the mixing of PANI/cellulose composite with silver powder. The addition of PANI structures in the silver ECA resulted in a significant increase of conductivity as well as provides enhanced mechanical properties. The work is well written and without a doubt deserves the publication after the following revision:

Major Remark

1 - The information about the silver flakes should be added: the SEM measurements should reveal the shape and size of silver nanostructures.

2 - The mutual distribution of PANi and Ag in created ECA should be characterized. The utilization of EDX mapping (in addition to provided SEM images) is suggested.

3 - Since the formation of the electric barrier between Ag and PANi can be expected, the utilization of voltammetry measurements is recommended. Obtained results can give a more precise explanation of the observed conductivity increase.

Minor Remark

1 - Detailed compositions for the 159 prepared ECAs with different compositions were given in Table

2 - The preparation of micro/nanostructures of conductive polymers has not been carefully described in the Introduction. The following articles are recommended: Long, C., et al. Advanced Functional Materials, 2014, 24(25), 3953-3961; Shi, Z., et al. RSC Advances, 2012, 2(3), 1040-1046; DÄ›kanovský, L., Advanced Functional Materials, 2019  29 (31) 1901880

3 - AFM – Fig. 2, please change the color of scale-bars

Reviewer 2 Report

This manuscript presents a kind of polyaniline@cellulose (PANI@CNs) nanowhiskers having a high aspect ratio and excellent solubility in 1,4-dioxane. It showed s superior conductivity and high mechanical properties. Especially, Cellulose nanowhiskers (CNs) were used for the most promising biomaterials. I think it's slightly contradictory. Although the obtained PANI@CNs suspension exhibits excellent dispersibility in 1,4-dioxane, I'm more concerned about 1,4-dioxane as solvent in terms of a biocompatibility. It, residue solvent, causes irritation of the eyes, nose and throat in humans for short periods of time. Exposure to large amounts can cause kidney and liver damage. Moreover, it had the lack in the novelty and did not present a break-through in this area (current open paper ref. Zhijun Shi et al RSC Advances 2012, 2, 1040–1046, Xiaodong Wu et al ACS Appl. Mater. Interfaces2014, 6, 23, 21078-21085, Taku Omura et al ACS Omega 2019, 4, 9446−9452). The entire logic was insufficient: the objective, necessity, and advantages of this study must be clarified in the manuscript with further explanation.

Reviewer 3 Report

This manuscript showed that the addition of cellulose nanowhiskers coated with polyaniline to silver flakes improves the conductivity and hardly changes it with mechanical bending. I think it is useful for printed electronics and can be published, There is a comments as follows:.

The authors use 1,4-dioxane, but dioxane has been pointed out to be potentially carcinogenic to humans, and its usage may be restricted in the future. I don't think it is suitable as a solvent for ECAs. I think dioxane is convenient for cellulose with many hydroxyl groups, but if you have the results of other water-soluble and safe solvents, please add them in the manuscript. If there is not, it is better to have a statement that it will be considered in the future

Round 2

Reviewer 2 Report

This revised manuscript provided the responses to reviewer' comments. So, I would like to recommend this manuscript for Nanomaterials.